# Synthesis and Structural Insight into poly(dimethylsiloxane)-*b*-poly(2-vinylpyridine) Copolymers

**DOI:** 10.3390/polym15214227

**Published:** 2023-10-25

**Authors:** Gkreti-Maria Manesi, Ioannis Moutsios, Dimitrios Moschovas, Georgios Papadopoulos, Christos Ntaras, Martin Rosenthal, Loic Vidal, Georgiy G. Ageev, Dimitri A. Ivanov, Apostolos Avgeropoulos

**Affiliations:** 1Department of Materials Science & Engineering, University of Ioannina, University Campus-Dourouti, 45110 Ioannina, Greece; gretimanesi@uoi.gr (G.-M.M.); imoutsios@uoi.gr (I.M.); dmoschov@uoi.gr (D.M.); gpap414@gmail.com (G.P.); ntaras@megaplast.gr (C.N.); 2Institut de Sciences des Matériaux de Mulhouse—IS2M, CNRS UMR7361, 15 Jean Starcky, 68057 Mulhouse, France; loic.vidal@uha.fr (L.V.); dimitri.ivanov@uha.fr (D.A.I.); 3Department of Chemistry, KU Leuven, Celestijnenlaan 200F, P.O. Box 2404, B-3001 Leuven, Belgium; martin.rosenthal@esrf.fr; 4Scientific Center for Genetics and Life Sciences, Sirius University of Science and Technology, 1 Olympic Ave., 354340 Sochi, Russia; ageev.gg@talantiuspeh.ru; 5Faculty of Chemistry, Lomonosov Moscow State University (MSU), GSP-1, 1-3 Leninskiye Gory, 119991 Moscow, Russia; 6Federal Research Center of Problems of Chemical Physics and Medicinal Chemistry RAS, Russian Academy of Sciences, Chernogolovka, 142432 Moscow, Russia

**Keywords:** high *χ* copolymers, anionic polymerization, chlorosilane chemistry, PDMS, P2VP, divergent T_g_s, self-assembly behavior, quaternization

## Abstract

In this study, the use of anionic polymerization for the synthesis of living poly(dimethylsiloxane) or PDMS-Li^+^, as well as poly(2-vinylpyridine) or P2VP-Li^+^ homopolymers, and the subsequent use of chlorosilane chemistry in order for the two blocks to be covalently joined leading to PDMS-*b*-P2VP copolymers is proposed. High vacuum manipulations enabled the synthesis of well-defined materials with different molecular weights (${\bar{Μ}}_{n}$, from 9.8 to 36.0 kg/mol) and volume fraction ratios (φ, from 0.15 to 0.67). The ${\bar{Μ}}_{n}$ values, dispersity indices, and composition were determined through membrane/vapor pressure osmometry (MO/VPO), size exclusion chromatography (SEC), and proton nuclear magnetic resonance spectroscopy (^1^H NMR), respectively, while the thermal transitions were determined via differential scanning calorimetry (DSC). The morphological characterization results suggested that for common composition ratios, lamellar, cylindrical, and spherical phases with domain periodicities ranging from approximately 15 to 39 nm are formed. A post-polymerization chemical modification reaction to quaternize the nitrogen atom in some of the P2VP monomeric units in the copolymer with the highest P2VP content, and the additional characterizations through ^1^H NMR, infrared spectroscopy, DSC, and contact angle are reported. The synthesis, characterization, and quaternization of the copolymer structure are important findings toward the preparation of functional materials with enhanced properties suitable for various nanotechnology applications.

## 1. Introduction

To obtain well-defined and stable nanostructures with extremely low dimensions, for nanotechnology purposes, highly immiscible copolymers should be designed [1,2,3,4,5]. Inherent properties, including thermal stability, etching selectivity, post-polymerization chemical modification capability, and high-throughput synthesis, are of great significance [1,2,3,4,5,6,7,8,9,10,11]. 

Silicon-containing copolymers have a leading role in various applications because they showcase the aforementioned characteristics [1,2]. This fact has motivated the scientific community towards the synthesis of various copolymer combinations, which include at least one inorganic segment. Several poly(dimethylsiloxane) or PDMS-based copolymers such as PS-*b-*PDMS [12,13], PDMS-*b-*P2VP [14], PDMS-*b-*P4VP [15], PDMS-*b-*PLA [16], PDMS-*b-*PMMA [17], P3HS-*b-*PDMS [18], PDMS-*b-*PEO [19], and PDMS-*b*-PTFEAs [20] [where PS: polystyrene, P2VP: poly(2-vinylpyridine), P4VP: poly(4-vinylpyridine), PLA: poly(lactid acid), PMMA: poly(methyl methacrylate), P3HS: poly(3-hydroxystyrene), PEO: poly(ethylene oxide and PTFEAs: poly(2,2,2-triflouroethyl acrylate)s] have been synthesized using living anionic, control radical polymerization techniques, ring-opening polymerization or azide-alkyne “click” reactions.

Poly(vinylpyridine)-based copolymers are highly versatile materials due to the chemical modification capability of the pyridinyl nitrogen atoms in the –ortho or –para positions [21,22,23,24,25,26,27]. Significant studies have shed light on the properties afforded after the reactions, which induce an electron-rich backbone and electron-poor pyridiniums for different quaternization degrees. It is straightforward that the properties and, therefore, the targeted application are dependent on the percentage of quaternization [21,22,23]. The tunable properties of the specific materials render them as possible candidates for various applications in diverse fields such as nanophotonic crystals (exquisite optical properties), stimuli-responsive materials (response to changes in pH), antimicrobial agents (interaction due to cationic nature), biosensors (detection of certain analytes due to electrical and/or fluorescence properties), and membranes (ion selectivity) [28,29,30,31].

Notwithstanding the fact that the advent of new synthetic approaches has enabled the fabrication of different systems, the synthesis of PDMS-*b-*P2VP remains largely unexplored. Their combination affords desirable properties due to the fact that a silicon-based flexible hydrophobic segment is combined with a stiff, amorphous block, which is capable of being modified, leading to completely different characteristics.

Regarding their preparation, Lee et al. [32] proposed certain synthetic protocols for the synthesis of linear diblock and triblock copolymers of the AB and ABA types (where A: P2VP and B: PDMS) that involved the use of an organolithium compound as initiator or a reagent with combined alkylating and acetal functional groups. Fragouli et al. [33] reported the synthesis of tetrablock quarterpolymers of the PS-*b*-PI-*b*-PDMS-*b*-P2VP [PI: poly(isoprene)] through anionic polymerization followed by coupling with a selective heterofunctional linking chlorosilane agent (chloromethylphenylethylenedimethylchlorosilane). Reversible addition-fragmentation chain transfer (RAFT) polymerization was employed quite recently by Hur et al. [34] for the synthesis of low molecular weight copolymers of the specific type. Only limited research works report the synthesis of the specific sequence, but its great potential is displayed in a work by Jeong et al. [14] in which the self-assembly behavior in thin films after exposure to different solvents, in a process widely established as solvent vapor annealing, was studied. Also, the self-assembly behavior of a PDMS-*b*-P2VP copolymer in thin films using pre-patterned substrates coated with different brushes was studied to evaluate the minimum value of line edge and/or width roughness [34]. Different research involved the fabrication of silicon oxide dots and the incorporation of gold nanoparticles in the polymeric matrix [35]. The bulk phase behavior prior to and after blending with 1-pyrenebutyric acid towards the formation of complexes through non-covalent interactions between the nitrogen of P2VP monomeric units and the butyric acid functional group as the temperature varied was examined by Shi et al. [36].

In our work, an alternative synthetic approach for the preparation of linear PDMS-*b*-P2VP copolymers is proposed. Living anionic polymerization was utilized for the synthesis of both living PDMS-Li^+^ and P2VP-Li^+^ homopolymers, which were then coupled using an appropriate reagent through chlorosilane chemistry. Combining the specific segments is rather challenging due to the quite different synthetic protocols followed during their polymerization. Restrictions related to electron affinity and weak nucleophilicity in conjunction with the strict purification protocols should be taken into consideration. The molecular characteristics of the involved segments, including the dispersity indices and total number average molecular weights, as well as the successful synthesis, were determined using size exclusion chromatography (SEC) and membrane/vapor pressure osmometry (MO/VPO). The mass as well as the volume fraction ratios were calculated using ^1^H NMR (proton nuclear magnetic resonance spectroscopy) experiments, while the thermal behavior was studied using differential scanning calorimetry (DSC). Two distinct glass transition temperatures corresponding to the PDMS and P2VP blocks were recorded, indicating the strong repelling forces between them. Lamellar, cylindrical, and spherical phases with dimensions ranging from approximately 15 to 39 nm were formed in bulk as indicated by morphological characterization through transmission electron microscopy (TEM) and small angle X-ray scattering (SAXS). Also, a post-polymerization chemical modification reaction towards the formation of a quaternized copolymer derivative was carried out on the sample with the highest P2VP content. The molecular, thermal, and wetting properties after the quaternization were determined to map any alternation induced due to the existence of a positive charge in some monomeric units of the P2VP segment. All samples are abbreviated as DVP-x (where x = 1, 2, 3) since three samples in total were synthesized.

## 2. Materials and Methods

### 2.1. Materials

The solvents (benzene, tetrahydrofuran, methanol, chloroform, and n-hexanes), monomers [1,1-diphenylethylene, 2-vinyl pyridine (2-VP) and hexamethylcyclotrisiloxane (D_3_)], coupling agent [dichlorodimethylsilane], initiators [*normal*-BuLi (*n*-BuLi), *secondary*-BuLi (*sec*-BuLi)], and drying agents (calcium hydride and triethylaluminum) have been provided by Sigma-Aldrich (Sigma-Aldrich Co., St. Louis, MO, USA), as well as the alkyl halide used in the quaternization (bromoethane) reaction. The purification procedures of all solvents and coupling agents were performed according to the demanding protocols of anionic polymerization, already described in the relative literature [37]. The apparatuses were meticulously rinsed with *n*-BuLi to eliminate any contaminants prior to any reaction. 2-VP was purified twice by calcium hydride prior to distillation to sodium mirror and finally to triethylaluminum [37,38]. D_3_ was dissolved in benzene, then was purified over calcium hydride as well as over living polystyrene lithium [PS^(−)^Li^(+)^ at least twice] chains and stored in pre-calibrated ampoules [12,37,39]. 

### 2.2. Synthesis Protocols

Synthesis of sample DVP-1 (see Table 1): The procedure is realized by polymerizing the hexamethylcyclotrisiloxane (6.0 g, 0.08 mol) with *sec*-BuLi (1.13 mmol) in the presence of non-polar solvent (benzene ~200 mL) followed by the addition of a polar solvent (tetrahydrofuran ~200 mL) in a 1:1 ratio after seven days at cryogenic conditions (−20 °C) as thoroughly elaborated in the literature [12,13]. Through this protocol, the conversion of monomer approaches 100% with minimum side reactions, which is critical for the effective coupling of the living PDMS homopolymer with dimethyldichlorosilane [(CH_3_)_2_SiCl_2_]. 

The molar ratio of the coupling agent towards the PDMS^−^Li^+^ is in significant excess (at least 500-fold, 0.57 mol) to ensure the substitution of only one chlorine atom [37,40,41]. The excess coupling agent was removed under high vacuum, and the intermediate PDMS-Si(CH_3_)_2_Cl product was washed with purified benzene, dissolved in tetrahydrofuran, and kept at a low temperature. The P2VP homopolymer was prepared in a different apparatus, which was meticulously rinsed with a bis phenyl hexyl lithium solution. 2-vinyl pyridine (6.0 g, 0.06 mol) and *sec*-BuLi reacted in the presence of tetrahydrofuran at −78 °C for 1 h using a nitrogen/isopropyl alcohol bath. The final step involved the coupling of the intermediate PDMS-Si(CH_3_)_2_Cl product and the living P2VP block. To control the instability of the materials at increased temperatures, the new apparatus with the attached vessels containing the two different segments is placed for approximately 15 min at −20 °C, and subsequently, the break-seal is ruptured towards the coupling of the macromolecular chains. The solution was left under stirring for approximately two weeks, where the gradual discoloration of the solution was apparent prior to the precipitation to cold stabilized methanol. Manipulating the monomers and/or initiator quantities but keeping coherent synthetic protocols, the remaining samples were synthesized. Note that after the completion of each synthetic step, aliquots were retrieved to determine the molecular characteristics and justify the successful reactions. Figure 1a summarizes all the reactions that took place for clarification reasons. Overall, three (3) samples were synthesized with different molecular characteristics and volume fractions depicted as DVP-1, DVP-2, and DVP-3, respectively (see Table 1).

Concerning the post-polymerization chemical modification reaction toward the formation of PDMS-*b*-[P2VP-*r*-poly(1-ethyl-2-vinylpyridinium bromide)], 1.0 g of sample DVP-3 (see Table 1) was vigorously stirred with an excess (with respect to the concentration of the pyridine units) of bromoethane (3.2 mL) under reflux for 16 h (Figure 1b) prior to purification, based on a previous study conducted by our research group [23].

### 2.3. Methods

A fully Integrated GPC System (PL-GPC 50) from Agilent Technologies (Agilent Technologies/Polymer Labs, St. Clara, CA, USA), with an isocratic pump (1.0 mL/min), a three-column [three columns in series (PLgel 5 mm Mixed-C, 300 × 7.5 mm) capacity oven (LabAlliance, New York, NY, USA) operating at 35 °C, refractive index (RI, Shodex RI-101, Munich, Germany) and ultraviolet absorbance (UV, SpectraSystem UV1000, Odense NV, USA) detectors was used. The eluent was stabilized (2% *v*/*v* triethylamine) THF, and the calibration took place using eight PS standards (M_p_: 1390 to 1,214,000 g/mol)]. 

Membrane Osmometry and Vapor pressure Osmometry [Gonotec Osmomat 070 at 45 °C or a Gonotec Osmomat 090 at 35 °C, respectively (Berlin, Germany)] using toluene as a solvent were used to determine the number-average molecular weight of the homopolymer precursors and the final copolymers.

^1^H NMR spectra were recorded in deuterated chloroform at 25 °C using Bruker AVANCE II spectrometers (Bruker GmbH, Berlin, Germany) operating at 250 MHz. All ^1^H NMR experiments in this work are reported in δ units, parts per million (ppm), and were measured relative to the signals for residual chloroform (7.26 ppm) in CDCl_3_. The CDCl_3_ used in this study contained no TMS as the internal reference. A UXNMR (Bruker) software (TopSpin v8.1) was used in the data analysis. The quaternized sample and the diblock copolymer precursor (DVP-3) were recorded in deuterated dimethyl sulfoxide (DMSO-d_6_) at 25 °C. 

DSC measurements were carried out on a *Q*20 TA instrument (TA Instruments Ltd., Leatherhead, UK) from −140 °C to 140 °C at a 10 °C/min heating rate. From the three cycles (two heating and one cooling) conducted, the presented thermographs corresponded to the second heating and were processed using Advantage v5.4.0 (TA instruments) software.

A Nicolet Nexus 670 (Wake Forest, NC, USA) infrared spectrometer equipped with a single horizontal golden gate attenuated total reflectance (ATR) cell was used for the FT-IR measurements. Spectra were recorded by averaging 64 scans between 4000 and 400 cm^−1^ with a resolution of 2 cm^−1^ under ambient conditions.

A contact angle instrument (OCA 25, DataPhysics Instruments GmbH, Filderstadt, Germany) was used to study the wetting properties. Samples were initially dissolved in chloroform (3 wt% solution) and were spin-cast onto silicon wafers [treated with piranha solution (sulfuric acid/hydrogen peroxide: 3/1)] at ambient conditions using 3700 rpm for 30 s to obtain films with ~50 nm thickness. Five separate measurements in five different regions were conducted, and the average value was calculated and presented. The deviation was ±2°.

A JEOL JEM 2100-HR, 200 KeV electron microscope (JEOL Ltd., Tokyo, Japan) was utilized for the TEM studies. Prior to TEM experiments, samples were cast from a dilute 5 wt% solution in chloroform, and the solvent evaporation was completed after 7 days. The obtained ultra-thin sections (ca. 50 nm thick) from a Leica EM UC7 ultramicrotome (Leica EM UC7 from Leica Microsystems, Wetzlar, Germany) at −140 °C were placed on 600 mesh copper grids.

Small angle X-ray scattering (SAXS) experiments were performed on a Xenocs Xeuss SAXS/WAXS (Holyoke, MA, USA) system equipped with a GeniX3D copper microfocus tube operating at 60 kV and 0.59 mA. All samples were placed in an evacuated chamber and illuminated with monochromatic X-rays in transmission geometry. The scattered intensity was recorded using a Dectris Pilatus 300k detector located 2.2 m downstream of the sample position. Small-angle X-ray scattering experiments were also conducted at the BM26 beamline of the ESRF (Grenoble).

## 3. Results and Discussion

Through anionic polymerization and chlorosilane chemistry, segments that are weak nucleophiles and not prone to sequential monomer addition can be synthesized. The linking of the living PDMS-Li^+^ homopolymer with the appropriate chlorosilane compound, followed by the reaction with the living P2VP-Li^+^ chains, enabled the synthesis of PDMS-*b*-P2VP copolymers with targeted molecular weight values and well-defined characteristics. This methodology constitutes a significant synthetic advance that can be further applied to segments with similar nucleophilicity-related issues. In this work, we demonstrate an additional route to synthesize the PDMS-*b*-P2VP copolymers that are desirable for diverse applications not only due to the high immiscibility they present but also because of the ability of the pyridine moiety to be chemically modified, leading to novel functional materials. 

The molecular characteristics were determined using size exclusion chromatography, membrane or vapor pressure osmometry, as well as proton nuclear magnetic resonance spectroscopy, and the thermal properties were studied with differential scanning calorimetry. Table 1 summarizes the molecular as well as the thermal characteristics of the copolymers (the samples are abbreviated as DVP) as directly calculated through the above-mentioned techniques.

As showcased in the relative SEC chromatograms, where the area from 0 up to 27 min is observed (Figure 2a), monomodal distributions were received for the final copolymers in the three different cases, indicating the absence of any undesired by-product prior to the self-assembly studies. The whole SEC chromatograms (including elution times from 0 up to 35 min) of the living PDMS-Li^+^ and P2VP-Li^+^ precursors together with the final copolymers in each case are presented in the Appendix A. 

The composition of the copolymers was calculated through the integration of the characteristic proton signals of each monomeric unit in the ^1^H NMR spectra (CDCl_3_). Specifically, at δ: 0.1–0.5 ppm, the chemical shifts are assigned to the six protons of the siloxane side methyl groups, while the peak area at δ: 8.2 ppm corresponds to the proton of the aromatic ring of the pyridine moiety as indicated in Figure 2b. Additionally, the peaks at 6.0–7.5 ppm are attributed to the three aromatic protons in the monomeric unit of P2VP. In all spectra, the targeted integrations through which the mass and, therefore, the volume fraction ratios can be calculated are presented as well. 

Concerning the thermal characterization results of the copolymers, DSC experiments indicated two distinct glass transition temperatures at values similar to those of the relative homopolymers. The significant difference between the two T_g_s, which is approximately 200 °C (lower than −121 °C and 100 °C for PDMS and P2VP, respectively), suggests the quite dissimilar properties of the components in terms of flexibility/stiffness between the two fractions. Any additional transitions are allocated to the melting and cold crystallization of the PDMS crystals. To calculate the degree of crystallinity of the PDMS crystals during melting in the DVP-1 and DVP-2, the enthalpy of fully crystallized PDMS was considered equal to 37.4 J/g based on the literature [42,43], leading to 19% and 25% of crystalline chains, respectively. The thermographs with the different thermal transitions are presented in Figure 2c. Sample DVP-3 did not showcase any melting and crystallization in the DSC thermograph (as evident in Figure 2c), which was associated with exclusively amorphous PDMS chains.

To establish a relationship between molecular characteristics and self-assembly behavior, TEM and SAXS experiments were conducted. The samples were cast in chloroform, which is more selective for the P2VP block if one takes into consideration the solubility parameters (chloroform: 18.7 MPa^1/2^, PDMS: 15.5 MPa^1/2^ and P2VP: 20.6 MPa^1/2^) [7,21,44] and the solvent was left to evaporate from the solutions at ambient conditions. The contrast obtained by silicon atoms in the PDMS block and carbon atoms in the P2VP moiety allowed for TEM observations without selective staining of the P2VP blocks with iodine. All as-cast samples were studied without being submitted to thermal annealing since copolymers showcase the highest repulsion at lower temperatures (the Flory–Huggins interaction parameter, *χ*, is inversely proportional with temperature). The results are summarized in Figure 3.

Sample DVP-1 formed alternating lamellae sheets as indicated by the 1D SAXS profile and the relative TEM micrograph after solution casting in chloroform [Figure 3a (blue line) and Figure 3b, respectively]. The black periodic regions correspond to the PDMS domains, while the white stripes correspond to the P2VP. The characteristic reflections at the relative *q* values of 1:2:3 further justified the real-space imaging results. The domain periodicity was calculated using the sharp primary reflection through Bragg’s law equal to 15.2 nm. The small dimensions obtained demonstrate the ability of the specific sequence to self-assemble despite the low degree of polymerization, and therefore, the strong immiscibility between the divergent components is verified.

The self-assembly studies on sample DVP-2 suggested the formation of hexagonal packed cylinders of the minority component or P2VP (white areas) inside the black PDMS matrix, as indicated in the relative TEM micrograph (Figure 3b). The reciprocal space results further confirmed the cylindrical morphology if one takes into consideration the characteristic reflections at the relative *q* values of 1:$\;\sqrt{4}$:$\;\sqrt{7}$:$\;\sqrt{9}$. The domain spacing was calculated through the principal reflection equal to 21.9 nm (Figure 3a, red line). 

Sample DVP-3 showcases the highest asymmetry between the molecular characteristics of the involved segments, which is also depicted in the morphological characterization results. Specifically, the copolymer adopted the spherical morphology as evident in the respective micrograph (Figure 3c), but no long-range order was observed, as indicated by the absence of sharp reflections in the corresponding 1D SAXS plot (Figure 3a, black line). Besides the lack of ordered domains, no solid results concerning the formed structure can be extracted due to the broad peaks, which do not allow for appropriate peak assignment (the arrows correspond to a *q* ratio of $\sqrt{2}$:$\;\sqrt{6}$:$\;\sqrt{14}$). The broad nanodomain boundaries are indicated by the weak intensity of peaks and reveal that the copolymer is not strongly segregated. The domain spacing was calculated to be approximately equal to 39 nm using the first permitted reflection.

Taking into consideration the experimental results, one can assume that the specific copolymer sequence adopts hexagonal and lamellar phases for common volume fraction ratios, while the highest asymmetry between the compositions of the components induced the formation of spherical morphology without long-range order. 

It has already been mentioned that P2VP-containing copolymers can be chemically modified after polymerization, yielding materials with desirable properties. To further make use of the pyridine moiety DVP-3 sample, as already mentioned in a previous section, it was submitted to a quaternization reaction due to its higher content of P2VP (*φ*_P2VP_ = 0.85) compared to the other two DVP samples, leading to PDMS-*b*-[P2VP-*r*-poly(1-ethyl-2-vinylpyridinium bromide)], which is depicted as DVP-3q. The sample was characterized at the molecular level through spectroscopic [^1^H NMR (DMSO-d_6_) and IR] techniques. Additionally, the thermal (DSC) and surface (CA) properties, in comparison with the ones observed in the pristine copolymer precursor, were accessed.

As far as the ^1^H NMR (DMSO-d_6_) experiments are concerned, the results showcased the successful quaternization of the P2VP moiety up to 20% if one considers the quantitative analysis between the quaternized and non-quaternized peaks, which is in good agreement with the reported results in the respective literature [23]. Specifically, the DVP-3q was dissolved in DMSO-d_6_ since quaternized pyridine blocks demonstrate better solubility in a more polar solvent. To better indicate the effect of quaternization on the ^1^H NMR spectrum, the pristine copolymer was recorded in DMSO-d_6_, and the comparative results are presented in Figure 4a. The emergence of an additional peak at approximately 8.85 ppm is allocated to the chemical shift of the proton in which the ethyl group alkylates the nitrogen atom (N^+^-CH). In the relative spectra, a decrease in the intensity of the phenyl proton peaks (at δ: 6.5–7.5 ppm) of the aromatic pyridine rings in the monomeric units and the appearance of a new peak with the noticeable intensity of the phenyl proton peak (at δ: 8.85 ppm) corresponding to the quaternized pyridine groups are observed. Furthermore, the signal at δ: 8.3 ppm, which is attributed to the N-C-H proton of P2VP that was a doublet prior to the modification, turned to a singlet after the completion of the reaction. Coherent results were also reported in the literature after quaternizing P2VP- and P4VP- containing copolymers. Also, broad peaks appeared in the region 0.5–1.0 ppm for the DVP-3q sample, which corresponds to the alkyl groups of the quaternization agent. As a complementary method to map any potential alternation on the absorption bands prior to and after the quaternization, infrared spectroscopy was used, and the comparative spectra are presented in Figure 4b. In the IR spectrum of the pristine copolymer, the absorption signals corresponding to the characteristic P2VP and PDMS groups are evident. In particular, the signals at 3000–3100 cm^−1^ indicate the presence of aromatic C-H stretching, and at 1400 and 1450 cm^−1^, the C=C stretching. The signals at 570 and 750 cm^−1^ correspond to the deformation of the aromatic ring. Also, at 1070–1340 cm^−1^, C-N stretching vibrations can be identified, while the absorption of the -C=N- groups is located at 1570–1660 cm^−1^. PDMS exhibit IR peaks at approximately 700–796 cm^−1^ due to the -CH_3_ rocking and Si-C stretching, at 1020–1074 cm^−1^ because of the Si-O-Si stretching, and finally at 1260–1259 cm^−1^, as well as at 2950–2960 cm^−1^, due to the -CH_3_ deformation and asymmetric -CH_3_ stretching in the Si-CH_3_ group, respectively. 

For the DVP-3q, the differentiation lies in the absorption bands located in the region between 1700 and 1450 cm^−1^. In particular, the band located at 1590 cm^−1^ is decreased due to the minimization of pyridine groups and the appearance of pyridinium cations [22]. A new band appeared at 1628 cm^−1^, which is attributed to the coordination bonds between nitrogen and bromine ions. In the Appendix A, a magnification of the IR spectra concerning the region ranging between 1800 to 1000 cm^−1^ is provided, and the aforementioned alternations are highlighted for better clarity.

Following, the thermal properties of the DVP-3q sample were studied through DSC, and the results are given in Figure 4c. DSC experiments on the modified copolymer indicated a slight increase in the T_g_ of the P2VP segments equal to 5.6 °C. The bromoethane quaternizes certain nitrogen atoms [~20% as indicated by the ^1^H NMR (DMSO-d_6_) experiments], forming the respective bromide. We speculate that the quaternization has an impact on the mobility of polymer chains and, therefore, impedes the diffusion of the polymer chains, which eventually leads to an increase in the T_g_. Relevant studies on quaternized P4VP homopolymers showcased similar results, strongly dependent on the modification extent [22]. The fact that random modified monomeric units are distributed throughout the P2VP matrix is unable to significantly affect the overall thermal behavior of the P2VP block. We expect that a higher quaternization degree may possibly lead to more significant changes in the thermal properties.

Finally, the wetting properties of both samples, pristine DVP-3 and modified DVP-3q in thin film states, were evaluated using water contact angle measurements. Quaternization of some pyridine moieties increased the overall polarity of the sample even though the yield did not exceed 20%. Specifically, the contact angle of the neat copolymer was calculated at approximately 95° (±2°), while the modified sample exhibited decreased water contact angle ca. 83° (±2°). The different surface properties of the DVP-3q sample are attributed to the secondary interactions (such as hydrogen bonds and dipole-dipole interactions) between the O-H bonds of water and the quaternized groups in some of the P2VP monomeric units. The results are presented in Figure 4d.

## 4. Conclusions

We propose a platform methodology to synthesize copolymers with segments presenting nucleophilicity-related issues through anionic polymerization. In particular, the living anionically synthesized homopolymer chains (i.e., PDMS-Li^+^ and P2VP-Li^+^) are covalently joined via chlorosilane chemistry leading to narrow dispersed PDMS-*b*-P2VP copolymers. Each step, during the different reactions, requires meticulous manipulations in terms of purification and vacuum quality to avoid the deactivation of reagents and the appearance of side reactions. The samples were characterized molecularly and thermally to determine the molecular weight values, dispersity indices, composition, and thermal transitions, including glass transition temperatures as well as melting and crystallization temperature of PDMS crystals, where evident. The solid-state properties of the copolymers were studied through TEM and SAXS, showcasing the self-assembly capacity of the specific system. The composition of the components allowed the formation of lamellar- and cylindrical structures with spacings of approximately 15 nm and 21 nm, respectively. Also, the highest asymmetry with respect to the segment’s content gave rise to a spherical morphology with a d value equal to 39 nm. The materials are potential candidates for nanotechnology applications due to the enhanced immiscibility of the constituents. The presence of nitrogen atoms in the P2VP moiety may possibly lead to the preparation of functional materials simply by conducting post-polymerization chemical modification reactions. As indicated by ^1^H NMR experiments, the alkylation of the nitrogen atom from the ethyl group induced additional peaks corresponding to the proton attached to the nitrogen atom (N^+^-CH) as well as to the ethyl groups of the modification compound. Complementary IR studies were carried out to detect any alternations due to the chemical modification reaction, even though the sensitivity is more limited compared to the ^1^H NMR experiments. Mild alternations in the thermal and wetting properties of the final modified copolymer were observed. The ability to tune the properties through post-polymerization chemical modification reactions is important for the design of new materials.

## Figures and Tables

**Figure 1 polymers-15-04227-f001:**
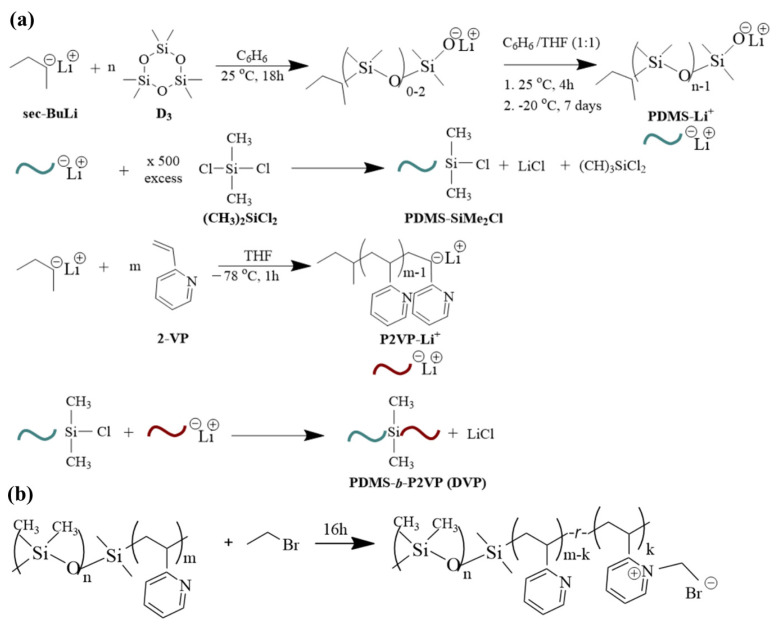
(**a**) Synthesis reactions for the preparation of PDMS-*b*-P2VP copolymers involving different steps, meaning the polymerization of D_3_ toward the formation of living PDMS chains, the coupling of PDMSLi^+^ with dichlorodimethylsilane, the polymerization of 2-VP toward the formation of living P2VP chains and finally the coupling of the chlorosilane terminated PDMS homopolymer with the active P2VP chains and (**b**) Post-polymerization chemical modification reaction toward the formation of PDMS-*b*-[P2VP-*r*-poly(1-ethyl-2-vinylpyridinium bromide)].

**Figure 2 polymers-15-04227-f002:**
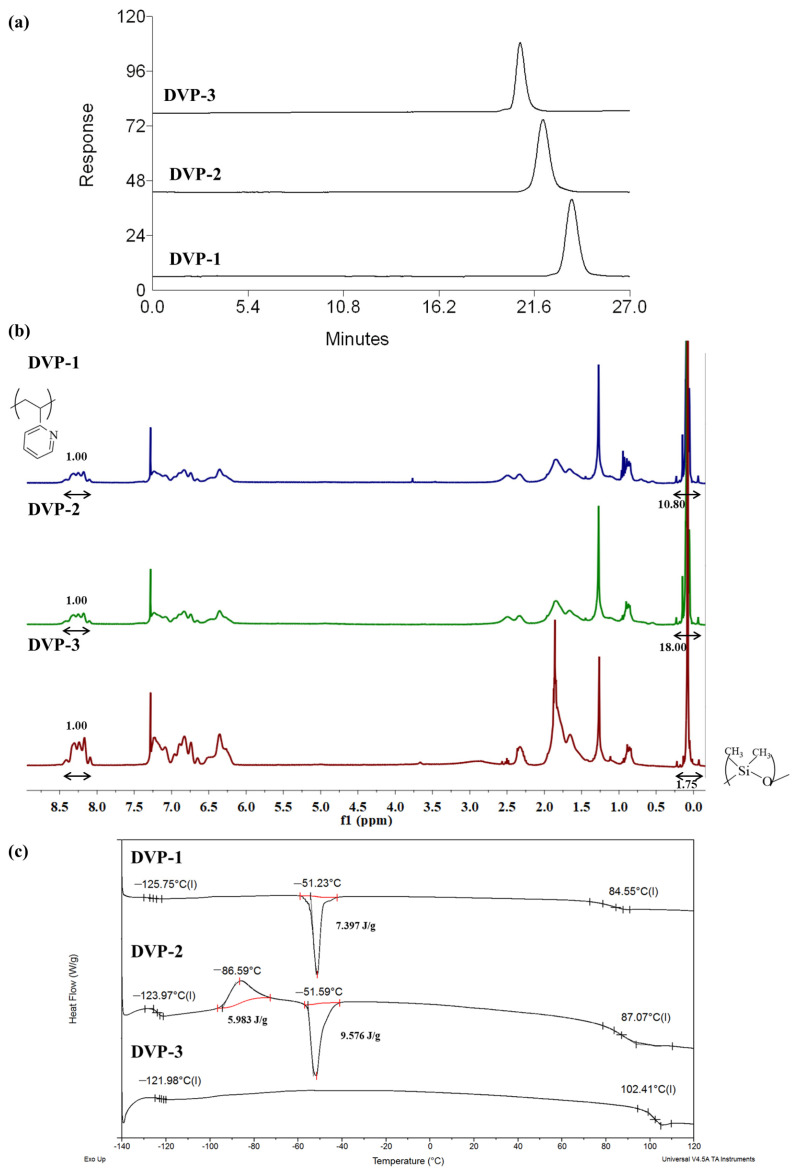
(**a**) SEC chromatograms, (**b**) ^1^H NMR (CDCl_3_) spectra, and (**c**) DSC thermograms of all synthesized copolymers as taken from the corresponding characterization techniques.

**Figure 3 polymers-15-04227-f003:**
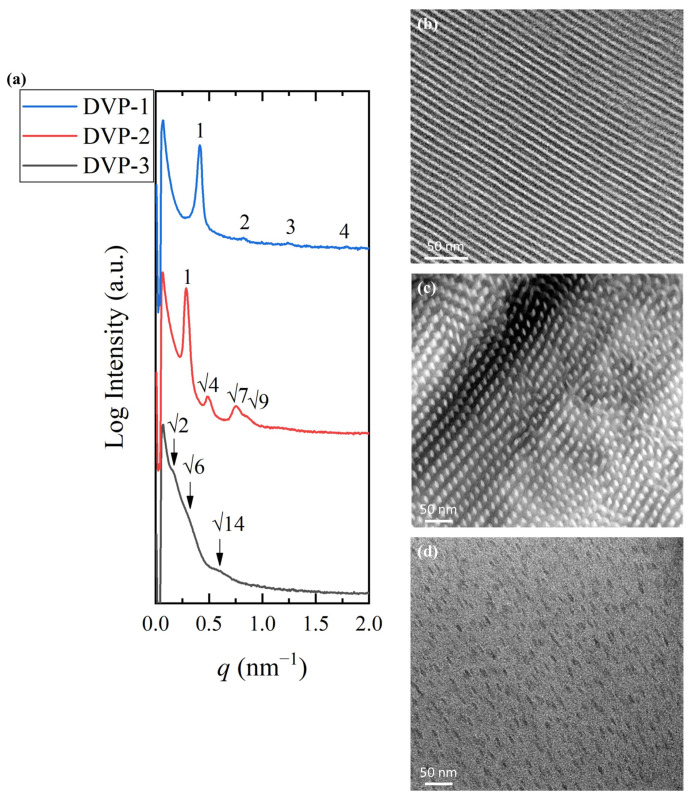
(**a**) 1D SAXS profiles of DVP-1 (blue) (the peak ratio of 1:2:3 indicates the formation of lamellar morphology), DVP-2 (red) (the peak ratio of 1:$\sqrt{4}$:$\sqrt{7}$:$\sqrt{9}$ indicates the formation of cylindrical morphology), and DVP-3 (black) (the peak ratio of 2:$\sqrt{6}$:$\sqrt{14}$ indicates the formation of spherical morphology). TEM micrographs of (**b**) DVP-1, (**c**) DVP-2, and (**d**) DVP-3 demonstrating alternating lamellae, hexagonal packed cylinders, and a spherical phase, respectively. The dark domains correspond to the PDMS and white/grey to the P2VP in all cases.

**Figure 4 polymers-15-04227-f004:**
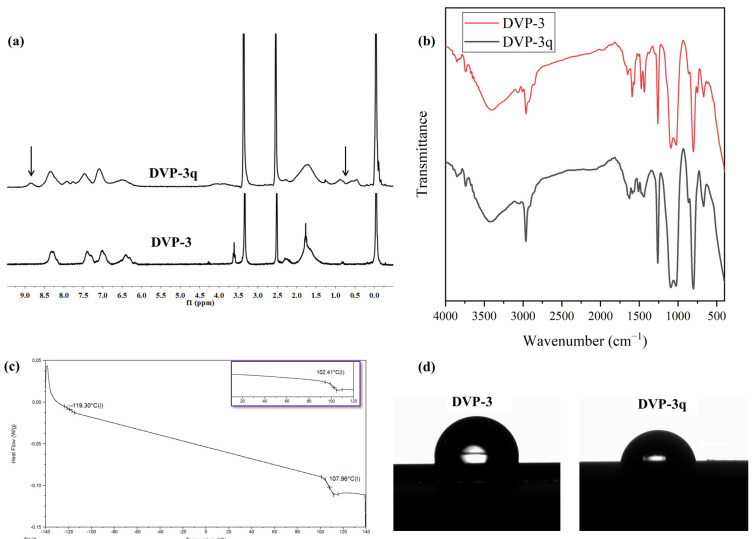
(**a**) Comparative ^1^H NMR (DMSO-d_6_) spectra of DVP-3 and DVP-3q. The arrows in the spectrum of DVP-3q indicate the emergence of the additional peaks which verify the successful quaternization, (**b**) Comparative FT-IR spectra of DVP-3 (red) and DVP-3q (black), (**c**) DSC thermograph of DVP-3q. The inlet in purple corresponds to the thermograph of the neat copolymer to indicate the temperature differentiation in the T_g_s before and after quaternization, (**d**) Water contact angle measurements of DVP-3 [95° (±2°)] and DVP-3q [83° (±2°)].

**Table 1 polymers-15-04227-t001:** Molecular and thermal characteristics of PDMS-*b*-P2VP copolymers, including number of average molecular weight (${\bar{Μ}}_{n}$), dispersity indices (Ð), mass fraction, and volume fraction of PDMS (*f_PDMS_* and *φ_PDMS_*_,_ respectively), as well as the glass transition temperatures of the chemically different segments.

Sample	${\bar{Μ}}_{n}$^PDMS (a)^ (g/mol) SEC/VPO or MO	${\bar{Μ}}_{n}$^P2VP (a)^ (g/mol) SEC/VPO or MO	${\bar{Μ}}_{n}$^TOTAL (a)^ (g/mol) SEC/VPO or MO	*Đ*^TOTAL (a)^ SEC	*f*_PDMS_ ^(b)^ *^1^H-NMR*	*φ*_PDMS_ ^(b)^	*T_g_*^PDMS^ ^(c)^ (°C)	*T_g_*^P2VP^ ^(c)^ (°C)
DVP-1	5300	4500	9800	1.06	0.55	0.54	−125.8	84.6
DVP-2	15,500	7000	22,500	1.05	0.68	0.67	−124.0	87.1
DVP-3	6000	30,000	36,000	1.04	0.16	0.15	−122.0	102.5

^(a)^ SEC in THF at 35 °C, VPO or MO in toluene at 45 °C or 35 °C, respectively. ^(b) 1^H NMR measurements in CDCl_3_ at 25 °C. The mass fraction ratio was calculated as follows: $f_{PDMS}=\frac{{\mathrm{integration}\;\mathrm{value}}_{0.1-0.5\;}}{\mathrm{contributing}\;\mathrm{protons}\;\left(\mathrm{or}\;6\right)}\times \;\mathrm{monomeric}\;\mathrm{unit}\;\mathrm{molecular}\;\mathrm{weight}\;\left(\mathrm{or}\;74\;g/\mathrm{mol}\right)$, $f_{P2VP}=\frac{{\mathrm{integration}\;\mathrm{value}}_{8.20}}{\mathrm{contributing}\;\mathrm{protons}\;\left(\mathrm{or}\;1\right)}\times \;\mathrm{monomeric}\;\mathrm{unit}\;\mathrm{molecular}\;\mathrm{weight}\;\left(\mathrm{or}\;105\;g/\mathrm{mol}\right)$. A characteristic example with respect to the calculation of mass fractions is provided in the Appendix A for comprehension reasons. Consequently the volume fraction is estimated through the following equations: $\varphi_{P2VP}=\frac{f_{P2VP}\rho_{PDMS}}{f_{P2VP}\rho_{PDMS}+\left(1-f_{P2VP}\right)\rho_{P2VP}}$, $\varphi_{PDMS}=\frac{f_{PDMS}\rho_{P2VP}}{f_{PDMS}\rho_{P2VP}+\left(1-f_{PDMS}\right)\rho_{PDMS}}$, where ρ_P2VP_ = 0.975 g/cm^3^ and ρ_PDMS_ = 0.970 g/cm^3^. ^(c)^ DSC experiments.

## Data Availability

The data presented in this study are available upon request from the corresponding author.

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
