# Peer review of "Synthesis and Structural Insight into poly(dimethylsiloxane)-b-poly(2-vinylpyridine) Copolymers"

_polymers, 2023, doi:10.3390/polym15214227_

Round 1

Reviewer 1 Report

1. The whole GPC profiles (Fig.2a) including those after 27min should be given, which could indicate solvent and especially some low molecular weight components possibly.

2. The living intermediates, PDMS-Li, P2VP-Li, are easy to be quenched and then their molecular weight could be also detected by GPC. Why not does it to give the corresponding data shown in Table 1?

3. The peaks should be explained at 6.0-7.5ppm of Fig.2b. The peaks at 0.1-0.5ppm assigned to Si-CH3, were integrated to calculate the ratio of PDMS. However, the internal standard, TMS in deuterium reagent for NMR, was appeared there as well. How to remove the interference? Why not adopt the integration data at 0.75-2.5ppm?

4. There should be the spectra of EDS, or elements mapping, corresponding to Fig.3b-d.

5. As a comparison before and after quaternization (Fig.4b), the spectra of 1H NMR should be given in the same deuterium reagent, for example, d-DMSO. Quaternization reaction should be elucidated briefly although revealed in reference.

6. The calculation formula of fpdms and fp2vp may be more clarified or explained.

7. The sentence maybe cause ambiguity in line 26-29, Page1.

Good.

Reviewer 2 Report

Comments are added.

it should be improved

Reviewer 3 Report

The authors present a systematic study on the synthesis and characterization of PDMS-b-P2VP block copolymers using anionic polymerization and chlorosilane chemistry. The work focuses on overcoming the synthetic challenges posed by the weak nucleophilicity of the blocks and describes the strict experimental protocols required. Molecular characterization confirms the successful synthesis of low dispersity copolymers. Phase behavior was studied, indicating the formation of lamellar, cylindrical and spherical nanostructures from self-assembly, with dimensions on the order of 15-40 nm. The P2VP block was partially quaternized and the modified polymer characterized. The manuscript is well-written and the results are interesting both fundamentally and for nanotechnology applications of high chi block copolymers. I recommend this manuscript for publication after minor revisions.

Concerns:

(1)            The abstract mentions that three copolymers were synthesized but their molecular characteristics are not summarized. This would help give readers a quick overview.

(2)            In the introduction, expand a bit more on the applications where these types of copolymers are useful.

(3)            In Fig. 4b, label the new peaks indicating quaternization.

(4)            Carefully proofread the manuscript to fix any minor grammar/typo issues.

Overall, this is a nice contribution on the synthesis and characterization of high immiscibility PDMS-b-P2VP copolymers. The results on overcoming synthetic challenges and the fundamental phase behavior studies significantly advance this field and I recommend publication after minor revisions.

Suggestions for improvement:

1)       In some places, the sentence structure is slightly awkward or can be tightened. Minor editing for flow and conciseness would help.

2)       Watch for any grammar issues with singular/plural, articles, prepositions etc. Proofreading will catch these.

3)       The writing could use more transition words between ideas to improve logical flow.

4)       Make sure verb tenses are consistent, especially when describing methods/results.

5)       The abstract could be more engaging by highlighting implications/importance.

Overall, the manuscript satisfies the requirements for proper academic English. With minor editing and proofreading, the language quality can be further polished.

Round 2

Reviewer 2 Report

Thank you very much for your suggestions and comments. The publication is now ready for publication.

Referee